# XNAS: Neural Architecture Search with Expert Advice

**Niv Nayman**[*]**, Asaf Noy**[*]**, Tal Ridnik**[*]**, Itamar Friedman, Rong Jin, Lihi Zelnik-Manor**

Machine Intelligence Technology, Alibaba Group
`{niv.nayman,asaf.noy,tal.ridnik,itamar.friedman,jinrong.jr,lihi.zelnik}`
`@alibaba-inc.com`

## Abstract

This paper introduces a novel optimization method for differential neural architecture search, based on the theory of prediction with expert advice. Its optimization criterion is well fitted for an architecture-selection, i.e., it minimizes the regret incurred by a sub-optimal selection of operations. Unlike previous search relaxations, that require hard pruning of architectures, our method is designed to dynamically wipe out inferior architectures and enhance superior ones. It achieves an optimal worst-case regret bound and suggests the use of multiple learning-rates, based on the amount of information carried by the backward gradients. Experiments show that our algorithm achieves a strong performance over several image classification datasets. Specifically, it obtains an error rate of $1.6\%$ for CIFAR-10, $23.9\%$ for ImageNet under mobile settings, and achieves state-of-the-art results on three additional datasets.

## 1 Introduction

In recent years tremendous efforts have been put into a manual design of high performance neural networks [22, 16, 40, 39]. An emerging alternative approach is replacing the manual design with automated Neural Architecture Search (**NAS**). NAS excels in finding architectures which yield state-of-the-art results. Earlier NAS works were based on reinforcement learning [55, 56], sequential optimization [24], and evolutionary algorithms [33], and required immense computational resources, sometimes demanding years of GPU compute time in order to output an architecture. More recent NAS methods reduce the search time significantly, e.g. via weight-sharing [30] or by a continuous relaxation of the space [25], making the search affordable and applicable to real problems.

While current NAS methods provide encouraging results, they still suffer from several shortcomings. For example, a large number of hyper-parameters that are not easy to tune, hard pruning decisions that are performed sub-optimally at once at the end of the search, and a weak theoretical understanding. This cultivates skepticism and criticism of the utility of NAS in general. Some recent works even suggest that current search methods are only slightly better than random search and further imply that some selection methods are not well principled and are basically random [23, 35].

To provide more principled methods, we view NAS as an online selection task, and rely on *Prediction with Experts Advice* (**PEA**) theory [4] for the selection. Our key contribution is the introduction of XNAS (eXperts Neural Architecture Search), an optimization method (section 2.2) that is well suited for optimizing inner architecture weights over a differentiable architecture search space (section 2.1). We propose a setup in which the experts represent inner neural operations and connections, whose dominance is specified by architecture weights.

---

[*]These authors contributed equally.

Our proposed method addresses the mentioned shortcomings of current NAS methods. For the mitigation of the hard pruning, we leverage the Exponentiated-Gradient (EG) algorithm [21], which favors sparse weight vectors to begin with, enhanced by a wipeout mechanism for dynamically pruning inferior experts during the search process. Additionally, the algorithm requires less hyper-parameters to be tuned (section 3.2.2), and the theory behind it further provides guidance for the choice of learning rates. Specifically, the algorithm avoids the decay of architecture weights [12], which is shown to promote selection of arbitrary architectures.

Additionally, XNAS features several desirable properties, such as achieving an optimal worst-case regret bound (section 3.1) and suggesting to assign different learning rates for different groups of experts. Considering an appropriate reward term, the algorithm is more robust to the initialization of the architecture weights and inherently enables the recovery of 'late bloomers', i.e., experts which may become effective only after a warm-up period (section 3.2.1). The wipeout mechanism allows the recovery of experts with a chance of being selected at the end of the process.

We compare XNAS to previous methods and demonstrate its properties and effectiveness over statistical and deterministic setups, as well as over 7 public datasets (section 4). It achieves state-of-the-art performance over 3 datasets, and top-NAS over rest, with significant improvements. For example, XNAS reaches $1.60\%$ error over CIFAR-10, more than $20\%$ improvement over existing NAS methods.

## 2 Proposed Approach

To lay out our approach we first reformulate the differentiable architecture search space of DARTS [25] in a way that enables direct optimization over the architecture weights. We then propose a novel optimizer that views NAS as an online selection task, and relies on PEA theory for the selection.

### 2.1 Neural Architecture Space

We start with a brief review of the PEA settings and then describe our view of the search space as separable PEA sub-spaces. This enables us to leverage PEA theory for NAS.

**PEA Settings.** PEA [4] refers to a sequential decision making framework, dealing with a decision maker, i.e. a forecaster, whose goal is to predict an unknown outcome sequence $\{y_t\}_{t=1}^T \in \mathcal{Y}$ while having access to a set of $N$ experts' advises, i.e. predictions. Denote the experts' predictions at time $t$ by $f_{t,1}, \ldots, f_{t,N} \in \mathcal{D}$, where $\mathcal{D}$ is the *decision space*, which we assume to be a convex subset of a vector space. Denote the forecaster's prediction $\{p_t\}_{t=1}^T \in \mathcal{D}$, and a non-negative loss function $\ell : \mathcal{D} \times \mathcal{Y} \to \mathbb{R}$. At each time step $t = 1, \ldots, T$, the forecaster observes $f_{t,1}, \ldots, f_{t,N}$ and predicts $p_t$. The forecaster and the experts suffer losses of $\ell_t(p_t) := \ell(p_t, y_t)$ and $\ell_t(f_{t,i}) := \ell(f_{t,i}, y_t)$ respectively.

**The Search Space Viewed as Separable PEA Sub-spaces.** We view the search space suggested by DARTS [25] as multiple separable sub-spaces of experts, as illustrated in Figure 1, described next. An architecture is built from replications of *normal* and *reduction cells* represented as a directed acyclic graph. Every node $x^{(j)}$ in this *super-graph* represents a feature map and each directed edge $(j, k)$ is associated with a forecaster, that predicts a feature map $p^{(j,k)} := p^{(j,k)}(x^{(j)})$ given the input $x^{(j)}$. Intermediate nodes are computed based on all of their predecessors: $x^{(k)} = \Sigma_{j<k} p^{(j,k)}$. The output of the cell is obtained by applying a reduction operation (e.g. concatenation) to the intermediate nodes. During the search stage, every forecaster combines $N$ experts' feature map predictions $\{f_i^{(j,k)}\}_{i=1}^N := \{f_i^{(j,k)}(x^{(j)})\}_{i=1}^N$ forming its own prediction,

$$p^{(j,k)} = \sum_{i=1}^N u_i^{(j,k)} f_i^{(j,k)} \quad ; \quad u_i^{(j,k)} = \frac{v_i^{(j,k)}}{\sum_{l=1}^N v_l^{(j,k)}} \quad ; \quad v_i^{(j,k)} \geq 0 \tag{1}$$

From now on, we will ignore the superscript indices $(j, k)$ for brevity. Each expert represents a neural operation, e.g. convolution or max pooling, associated with network weights $w_i$, that receives an input $x_t$ at time $t$, and outputs a feature map prediction. Thus a time index is attached, as each prediction $f_{t,i} := f_i(x_t)$ is associated with updated weights $w_{t,i}$.

Our architecture search approach is composed of two stages. In the *search stage*, the weights $w_{t,i}$ and $v_{t,i}$ are alternately optimized as described in section 2.2; then, in the *discretization stage*, a discrete child architecture is obtained as explained next.

**The Discretization Stage.** Once the architecture weights are optimized, the final discrete neural architecture is obtained by performing the following discretization stage, adopted from [25]: Firstly, the strongest two predecessor edges are retained for each intermediate node. The strength of an edge is defined as $\max_i u_i^{(k,j)}$. Lastly, every forecaster is replaced by the corresponding strongest expert.

## 2.2 XNAS: eXperts Neural Architecture Search

The differential space, described in section 2.1, enables direct optimization over the architecture weights via gradient-descent based techniques. Previous methods adopted generic optimizers commonly used for training the network weights. For example [25, 49, 6, 3] used adam [20], and [29] used SGD with momentum. While those optimizers excel in joint minimization of neural network losses when applied to network weights, NAS is a essentially a *selection* task, aiming to select a subset of experts out of a superset. The experts weights form a convex combination, as they compete over a forecaster's attention.

We argue that a generic alternate optimization of network weights and architecture weights, as suggested in previous works, e.g. [17, 25], is not suitable for the unique structure of the architecture space. Hence, we design a tailor-made optimizer for this task, inspired by PEA theory. In order to evaluate experts' performance, a loss is to be associated with each expert. However, an explicit loss is not assigned to each expert, as opposed to a back-propagated loss gradient. Therefore, we base our algorithm on a version of the Exponentiated-Gradient (EG) algorithm adapted for the NAS space. EG algorithms favor sparse weight vectors [21], thus fit well to online selection problems.

We introduce the XNAS (eXperts Neural Architecture Search) algorithm, outlined in Algorithm 1 for a single forecaster. XNAS alternates between optimizing the network weights $\boldsymbol{w}$ and the architecture weights $\boldsymbol{v}$ in a designated manner. After updating $\boldsymbol{w}$ (line 4) by descending the train loss $\ell_{train}$, each forecaster makes predictions over the validation set, based on the associated mixture of experts (line 5), as they propagate forward through the super-graph, incurring a validation loss $\ell_{val}$. Then $\boldsymbol{v}$ are updated according to the EG rule (lines 8-10) with respect to the validation loss, whose gradients are backward propagated to every forecaster through the super-graph (line 6). Next, the optimizer wipes-out weak experts (lines 12-13), effectively assigning their weights to the remaining ones. The exponential update terms, i.e. rewards, are determined by the projection of the loss gradient on the experts' predictions: $R_{t,i} = -\nabla_{p_t}\ell_{val}(p_t) \cdot f_{t,i}$. In section 3.1 we relate to the equivalence of this reward to the one associated with *policy gradient* search [45] applied to NAS.

---

**Algorithm 1** XNAS for a single forecaster

1: **Input**: The learning rate $\eta$, Reward bound $\mathcal{L}$ , Experts predictions $\{f_{t,i}\}_{i=1}^{N} \; \forall t = 1, \dots, T$
2: **Init**: $I_0 = \{1, \dots, N\}$, $v_{0,i} \leftarrow 1, \; \forall i \in I_0$
3: **for** rounds $t = 1, \dots, T$ **do**
4: &emsp; Update $\boldsymbol{\omega}$ by descending $\nabla_{\boldsymbol{\omega}}\ell_{\text{train}}(\boldsymbol{\omega}, \boldsymbol{v})$
5: &emsp; $p_t \leftarrow \dfrac{\sum_{i \in I_{t-1}} v_{t-1,i} \cdot f_{t-1,i}}{\sum_{i \in I_{t-1}} v_{t-1,i}}$ &emsp; #Predict
6: &emsp; $\{$loss gradient revealed: $\nabla_{p_t}\ell_{\text{val}}(p_t)\}$
7: &emsp; **for** $i \in I_{t-1}$ **do**
8: &emsp;&emsp; $R_{t,i} = -\nabla_{p_t}\ell_{\text{val}}(p_t) \cdot f_{t,i}$ &emsp; #Rewards
9: &emsp;&emsp; $R_{t,i} \leftarrow \min\{\max\{R_{t,i}, -\mathcal{L}\}, \mathcal{L}\}$
10: &emsp;&emsp; $v_{t,i} \leftarrow v_{t-1,i} \cdot \exp\{\eta R_{t,i}\}$ &emsp; #EG step
11: &emsp; **end for**
12: &emsp; $\theta_t \leftarrow \max_{i \in I_{t-1}}\{v_{t,i}\} \cdot \exp\{-2\eta\mathcal{L}(T-t)\}$
13: &emsp; $I_t \leftarrow I_{t-1} \setminus \{i \mid v_{t,i} < \theta_t\}$ &emsp; #Wipeout
14: **end for**

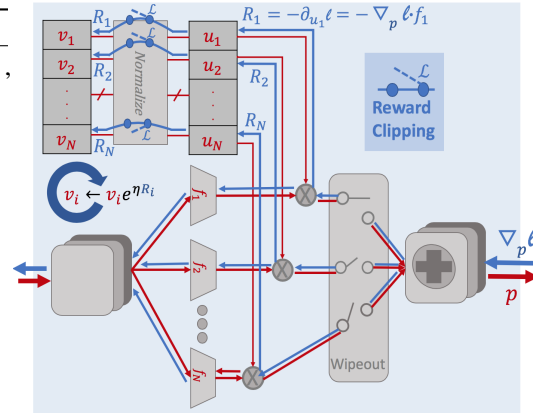

Figure 1: A visual illustration of a single forcaster, equipped with XNAS forward and backward propagation. Forward and backward passes are presented as red and blue arrows respectively.

The purpose of the wipeout step is threefold. First, the removal of weak experts consistently smooths the process towards selecting a final architecture at the descretization stage described in section 2.1. Thus it mitigates the harsh final pruning of previous methods, which results in a *relaxation bias* addressed in [50, 29]. Second, it dynamically reduces the number of network weights, thus simplifying the optimization problem, avoiding over-fitting and allowing it to converge to a better solution. Last, it speeds-up the architecture search, as the number of graph computations decreases with the removal of experts.

## 3   Analysis and Discussion

### 3.1   Theoretical Analysis

In this section we analyse the performance of the proposed algorithm. For this purpose, we introduce the *regret* as a performance measure for NAS algorithms. Showing that the wipeout mechanism cannot eliminate the best expert (lemma 1), we provide theoretical guarantees for XNAS with respect to that measure (theorem 1). Proofs appear in Section 7 of the supplementary material for brevity. Relaying on the theoretical analysis, we extract practical instructions with regard to the choice of multiple learning rates. Finally we briefly discuss the equivalence of our reward to the one considered by the policy gradient approach applied to NAS.

**The Regret as a Performance Measure.** Denote the regret, the cumulative losses of the forecaster and of the $i$th expert at time $t$ by,

$$\text{Regret}_t = L_t - \min_{i=1,...N} L_{t,i} \quad ; \quad L_t = \sum_{s=1}^{t} \ell_s(p_s) \quad ; \quad L_{t,i} = \sum_{s=1}^{t} \ell_s(f_{s,i}) \tag{2}$$

respectively. The regret measures how much the forecaster regrets not following the advice of the best expert in hindsight. This criterion suits our setup as we optimize a mixture of experts and select the best one by the end of the process.

In classical learning theory, statistical properties of the underlying process may be estimated on the basis of stationarity assumptions over the sequence of past observations. Thus effective prediction rules can be derived from these estimates [4]. However, NAS methods that alternately learn the architecture weights and train the network weights are highly non-stationary. In PEA theory, no statistical assumptions are made, as "simplicity is a merit" [14], and worst-case bounds are derived for the forecaster's performance. We obtain such bounds for the wipeout mechanism and the regret.

**A Safe Wipeout.** In XNAS (line 12), by the choice of the wipeout thresholds, experts with no chance of taking the lead by the end of the search are wiped out along the process. In a worse-case setup, a single incorrect wipeout might result in a large regret, i.e., linear in the number of steps $T$, due to a loss gap at each consecutive step. The following lemma assures that this cannot happen,

**Lemma 1.** *In XNAS, the optimal expert in hindsight cannot be wiped out.*

The wipeout effectively transfers the attention to leading experts. Define the *wipeout factor* and the *aggregated wipeout factor* as $\Gamma_t := 1 + \frac{\sum_{i \in I_{t-1} \setminus I_t} v_{t,i}}{\sum_{i \in I_t} v_{t,i}}$ and $\gamma_T := \prod_{t=1}^{T} \Gamma_t$, respectively.

**Lemma 2.** *The aggregated wipeout factor satisfies $1 \leq \gamma_T < N$.*

Equipped with both lemmas, we show that the wipeout may improve the EG regret bound for certain reward sequences.

**Regret Bounds.** Our main theorem guarantees an upper bound for the regret,

**Theorem 1** (XNAS Regret Bound). *The regret of the XNAS algorithm 1, with $N$ experts and learning rate $\eta$, incurring a sequence of $T$ non-negative convex losses of $\mathcal{L}$-bounded rewards, satisfies,*

$$\text{Regret}_T \leq \frac{\eta T \mathcal{L}^2}{2} + \frac{1}{\eta} \ln N - \frac{1}{\eta} \ln \gamma_T \tag{3}$$

As an input parameter of XNAS, the learning rate $\eta$ cannot be determined based on the value of $\gamma_T$, since the later depends on the data sequence. Choosing the minimizer $\eta^*$ of the first two terms of (3)

fully known in advance, yields the following tight upper bound,

$$\eta^* = \sqrt{\frac{2\ln N}{T\mathcal{L}^2}} \quad ; \quad \text{Regret}_T \leq \mathcal{L}\sqrt{2T\ln N}\left(1 - \frac{1}{2}\frac{\ln\gamma_T}{\ln N}\right) \tag{4}$$

The regret upper bound of XNAS is tight, as the lower bound can be shown to be of $\Omega(\mathcal{L}\sqrt{T\ln N})$ [13]. In addition, the wipeout related term reduces the regret in an amount which depends on the data sequences through $\gamma_T$, as it effectively contributes the attention of weak experts to the leading ones. For comparison, under the same assumptions, the worst-case regret bound of gradient-descent is of $O(\mathcal{L}\sqrt{TN})$ [14], while the one of Adam is linear in $T$ [34]. An illustration of the relationship between the regret and the rate of correct expert selection appears in section 8.3 of the supplementary material, where XNAS is shown to achieve a better regret compared to a generic optimizer.

**Multiple Learning Rates.** Equation 4 connects the optimal theoretical learning rate $\eta^*$ with the number of steps $T$, which is also the number of gradient feedbacks received by the experts. Since forecasters weights are being replicated among different cells, the number of feedbacks is different for normal and reduction cells (section 2.1). Explicitly, $T_c = d \times \mathcal{E} \times r_c$, where $T_c, d, \mathcal{E}, r_c$ are the effective horizon $T$, the validation set size, the number of epochs and the number of replications for cell type $c$ respectively. We adopt the usage of multiple learning rates $\eta^*_c$ in our experiments as upper bounds on the learning rates for minimizing the upper bound of the regret.

**The Connection to Policy Gradient.** We conclude this section by pointing out an interesting connection between *policy gradient* in NAS [55] and PEA theory. We refer to the PEA based reward term in line 8 of algorithm 1. This reward has been shown by [50] to be the same effective reward of a policy gradient method applied to the common NAS optimization criterion [55, 56, 30]. More precisely, consider the case where instead of mixing the experts' predictions, the forecaster is to sample a single expert at each step with probability $u_{t,i}$, specified in (1). Then the effective reward associated with the policy gradient will be exactly the derived PEA reward, $R_{t,i} = -\nabla_{p_t}\ell_t(p_t) \cdot f_{t,i}$. XNAS optimizes with respect to the same reward, while avoiding the sampling inefficiency associated with policy gradient methods.

### 3.2 Key Properties and Discussion

In this section we discuss some of the key properties of XNAS. For each of this properties we provide supporting derivations, illustrations and demonstrations appearing in section 8 of the supplementary material for brevity.

#### 3.2.1 The Recovery of Late Bloomers and Robustness to Initialization

In this section we point out a key difference between our proposed update rule and the one used in previous works. We refer to Gradient Descent (GD) with softmax as updating the parameters $\alpha_{t,i} = \ln v_{t,i}$, by descending $\nabla_{\alpha_{t,i}}\ell_t(p_t)$ respectively. Variants of GD with softmax, as used in [25] to optimize the architecture weights, suppress operations that are weak at the initial iterations, making it more difficult for them to "bloom" and increase their weights. This could be problematic, e.g. in the two following cases. First, consider the best expert starting with a poor weight which gradually rises. This could be the case when an expert representing a parameterized operation (e.g. a convolutional layer) competes with an unparameterized one (e.g. a pooling layer), as the first requires some period for training, as stated by [29, 17]. Second, consider a noisy setup, where the best expert in hindsight could receive some hard penalties before other inferior experts do. In NAS we deal with stochastic settings associated with the training data.

We inspect the update term of GD with softmax,

$$v_{t,i} = \exp\{\alpha_{t,i}\} = \exp\{\alpha_{t-1,i} - \eta\partial_{\alpha_{t-1,i}}\ell_t(p_t)\} = v_{t-1,i} \cdot \exp\{-\eta\partial_{\alpha_{t-1,i}}\ell_t(p_t)\} \tag{5}$$

Hence, the effective reward in this case is,

$$\tilde{R}_{t,i} := -\partial_{\alpha_{t-1,i}}\ell_t(p_t) = -\nabla_{p_t}\ell_t(p_t) \cdot u_{t-1,i}(f_{t,i} - p_t) \tag{6}$$

See derivations in section 8.4. The linear dependence on the expert's weight $u_{t-1,i}$ in (6) implies that GD with softmax makes it harder for an expert whose weight is weak at some point to recover and become dominant later on, as the associated rewards are attenuated by the weak expert's weight.

XNAS mitigates this undesirable behavior. Since for XNAS the update term (8) depends on the architecture weights only indirectly, i.e. through the prediction, the recovery of *late bloomers* is not discouraged, as demonstrated in section 8.1 of the supplementary material. From the very same reasons, XNAS is more robust to the initialization scheme compared to GD with softmax and its variants, as demonstrated in section 8.2 of the supplementary material. These advantages make XNAS more suitable for the NAS setup.

Note that while the XNAS enables the recovery of experts with badly initialized weights or with delayed rewards, the wipeout mechanism prevents inferior operations that start blooming too late from interfering, by eliminating experts with no chance of leading at the end.

**Wipeout Factor.** As mentioned in section 2.2, the wipeout mechanism contributes to both optimization process and search duration. A further reduction in duration can be achieved when the wipe-out threshold in line 12 of Algorithm 1 is relaxed with a parameter $0 < \zeta \leq 1$, being replaced by $\theta_t \leftarrow \max_{i \in I_{t-1}} \{v_{t,i}\} \cdot \exp\{-2\eta \mathcal{L}(T-t) \cdot \zeta\}$. This will lead to a faster convergence to a single architecture, with the price of a violation of the theoretical regret. As worst-case bounds tend to be over pessimistic, optimizing over $\zeta$ could lead to improved results. We leave that for future work.

### 3.2.2   Fewer Hyper Parameters

The view of the differentiable NAS problem as an optimization problem solved by variants of GD, e.g. Adam, introduces some common techniques for such schemes along with their corresponding hyper-parameters. Tuning these complicates the search process - the fewer hyper-parameters the better. We next discuss how XNAS simplifies and reduces the number of hyper-parameters.

**Theoretically Derived Learning Rates.** The determination of the learning rate has a significant impact on the convergence of optimization algorithms. Various scheduling schemes come up, e.g. [26, 38], as the later additionally suggests a way for obtaining an empirical upper bound on the learning rate. In section 3.1, multiple learning rates $\eta_c^*$ are suggested for minimizing the regret bound (4), as $c \in \{N, R\}$ represents normal and reduction cells respectively. For example, for CIFAR10 with 50%:50% train-validation split, 50 search epochs, gradient clipping of 1, 6 normal cells and 2 reduction cells both of 8 experts for each forecaster, (4) yields $\eta_N^* =$7.5e-4 and $\eta_R^* =$1.3e-3.

**Remark 1.** *Note that the proposed learning rates minimize an upper bound of the regret (4) in the case of no wipeout, i.e. the worst case, as the extent of the wipeout cannot be known in advance. Hence the proposed learning rate provides an upper bound on the optimal learning rates and can be further fine-tuned.*

**No Weight Decay.** Another common technique involving hyper-parameters is weight decay, which has no place in the theory behind XNAS. We claim that the obviation of weight decay by XNAS makes sense. Regularization techniques, such as weight decay, reduce overfitting of over-parametrized models when applied to these parameters [12]. No such effect is incurred when applying weight decay on the architecture parameters as they do not play the same role as the trained network parameters $w$. Instead, weight decay encourages uniform dense solutions, as demonstrated in Figure 2, where the mean normalized entropy increases with the weight decay coefficient. The calculation of the mean normalized entropy is detailed in section

Figure 2: Mean normalized entropy vs weight decay. The red dot refers to DARTS' settings.

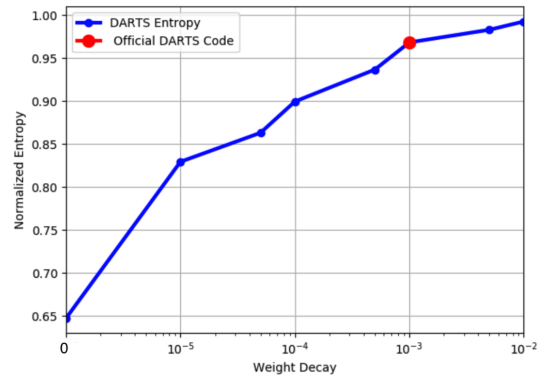

8.5 of the supplementary material. This observation could be associated with the suggestion of recent works [23, 35] that current search methods are only slightly better than random search. The density of results in a harder degradation in performance once *discretization stage* occurs (section 2.1), hence sparse solutions are much preferred over dense ones.

**No Momentum.** The theory behind XNAS obviates momentum [31] and ADAM's exponentially decay rates [20]. Since momentum requires more state variables and more computations, the resulting XNAS optimizer turns out to be simpler, faster and with a smaller memory footprint, compared to commonly used optimizers for NAS, e.g. ADAM [25, 49, 6, 3] and SGD with momentum [29].

# 4    Experiments and Results

In this section we will test XNAS on common image classification benchmarks, and show its effectiveness compared to the other state-of-the-art models.

We used the CIFAR-10 dataset for the main search and evaluation phase. In addition, using the cell found on CIFAR-10 we did transferability experiments on the well-known benchmarks ImageNet, CIFAR-100, SVHN, Fashion-MNIST, Freiburg and CINIC10.

## 4.1    Architecture Search on CIFAR-10

Using XNAS, we searched on CIFAR-10 in a small parent network for convolutional cells. Then we built a larger network by stacking the learned cells, trained it on CIFAR-10 and compared the results against other NAS methods.

We created the parent network by stacking $8$ cells with $4$ ordered nodes, each of which connected via forecasters to all previous nodes in the cell and also to the two previous cells outputs. Each forecaster contains seven operations: 3x3 and 5x5 separable and dilated separable convolutions, 3x3 max-pooling, 3x3 average-pooling and an identity. A cells output is a concatenation of the outputs of the four cells nodes.

The search phase lasts up to $50$ epochs. We use the first-order approximation [25], relating to $v$ and $\omega$ as independent parameters which can be optimized separately. The train set is divided into two parts of equal sizes: one is used for training the operations weights $\omega$ and the other for training the architecture weights $v$, both with respect to the cross entropy loss. With a batch size of 96, one epoch takes $8.5$ minutes in average on a single GPU[2] , summing up to 7 hours in total for a single search. Most of the speedup over related differentiable NAS methods is achieved by the quick EG updates and the rest by the wipeout. Figure 9 from the supplementary material shows a searched normal and reduction cells that will be used in the following evaluations.

## 4.2    CIFAR-10 Evaluation Results

We built the evaluation network by stacking 20 cells: 18 normal cells and 2 reduction cells. The reduction cells are placed after $1/3$ and $2/3$ of the network. After each reduction cell we double the amount of channels in the network. We trained the network for $1500$ epochs using a batch size of 96 and SGD optimizer with nesterov-momentum. Our learning rate regime was composed of $5$ cycles of power cosine annealing learning rate [17], with amplitude decay factor of $0.5$ per cycle. For regularization we used cutout [9], scheduled drop-path [22], auxiliary towers [39], label smoothing [40] AutoAugment [7] and weight decay. To understand the effect of the network size on the final accuracy, we chose to test three architecture configurations, XNAS-Small, XNAS-Medium and XNAS-Large, with $36$, $44$ and $50$ initial network channels respectively. Table 1 shows the performance of XNAS architectures compared to other state-of-the-art NAS methods. XNAS' smallest network variant, XNAS-Small, outperforms previous NAS methods by a large margin, with a minimal error rate of $1.81\%$ measured across 5 seeds with error rate mean and standard deviation of $1.91\% \pm 0.07\%$. Training the same model with the settings of [25] yields $2.48\%$ error rate. Our largest network variant, XNAS-Large, is the second highest reported score on CIFAR-10 (without any additional pre-train data), while having $4$ times less parameters than the top one [7][3]. In addition, XNAS is among the fastest NAS methods

| CIFAR-10 Architecture | Test error | Params (M) | Search cost |
|---|---|---|---|
| AutoAugment [7] | 1.48 | 26 | 20 |
| NAONet-WS [27] | 3.53 | 2.5 | 0.3 |
| PNAS [24] | 3.41 | 3.2 | 150 |
| Amoeba-A [33] | 3.34 | 3.2 | 3150 |
| DSO-NAS [53] | 2.95 | 3 | 1 |
| DARTS(1nd) [25] | 2.94 | 2.9 | 0.4 |
| ENAS [30] | 2.89 | 4.6 | 0.5 |
| SNAS [50] | 2.85 | 2.8 | 1.5 |
| DARTS(2nd) [25] | 2.67 | 3.4 | 1 |
| NASNet-A [56] | 2.65 | 3.3 | 1800 |
| Amoeba-B [33] | 2.55 | 2.8 | 3150 |
| PDARTS [6] | 2.50 | 3.4 | 0.3 |
| NAONet [27] | 2.11 | 128 | 200 |
| ProxylessNAS [2] | 2.08 | 5.7 | 10 |
| ASAP [29] | 1.68 | 6.0 | 0.2 |
| SharpDarts [17] | 1.93 | 3.6 | 0.8 |
| XNAS-Small | 1.81 | 3.7 | 0.3 |
| XNAS-Medium | 1.73 | 5.6 | 0.3 |
| XNAS-Large | **1.60** | 7.2 | 0.3 |

Table 1: Classification errors of XNAS compared to state-of-the-art methods on CIFAR-10. The Search cost is measured in GPU days. Test error refers to top-1 test error (%). XNAS Small, Medium and Large refer to network architectures with XNAS cells and 36, 44 and 50 initial channels respectively.

## 4.3 Transferability Evaluation

Using the cell found by XNAS search on CIFAR-10, we preformed transferability tests on 6 popular classification benchmarks: ImageNet, CIFAR-100, Fashion-MNIST, SVHN, Freiburg and CINIC10.

**ImageNet Results.** Our ImageNet network was composed of 14 stacked XNAS cells, with two initial stem cells for downscaling. We used 46 initial channels so the total number of network FLOPs is below 600[M][3], similar to other ImageNet architectures with small computation regime [48]. We trained the network for 250 epochs with one cycle of power cosine learning rate and a nesterov-momentum optimizer. The results are presented in Table 2. We can see from Table 2 that XNAS transferability results on ImageNet are highly competitive, outperforming all previous NAS cells.

| **ImageNet** Architecture | Test error | Params (M) | Search cost |
|---|---|---|---|
| SNAS [50] | 27.3 | 4.3 | 1.5 |
| ASAP [29] | 24.4 | 5.1 | 0.2 |
| DARTS [25] | 26.7 | 4.9 | 1 |
| NASNet-A [56] | 26.0 | 5.3 | 1800 |
| PNAS [24] | 25.8 | 5.1 | 150 |
| Amoeba-A [33] | 25.5 | 5.1 | 3150 |
| RandWire [48] | 25.3 | 5.6 | 0 |
| SharpDarts [17] | 25.1 | 4.9 | 0.8 |
| Amoeba-C [33] | 24.3 | 6.4 | 3150 |
| XNAS | **23.9** | 5.2 | 0.3 |

Table 2: Transferability classification error of XNAS, compared to top NAS cells, on ImageNet. Test error refers to top-1 test error (%). Search cost is measured in GPU days.

**Additional Results.** We further tested XNAS transferability abilities on 5 smaller datasets: CIFAR-100 [42], Fashion-MNIST [47], SVHN [28], Freiburg [19] and CINIC10 [8]. We chose to use the XNAS-Small architecture, with similar training scheme to the one described in section 4.2. Table 3 shows the performance of our model compared to NAS methods. We can see that XNAS cell excels on the datasets tested. On CIFAR-100 it surpasses the next top cell by 1%, achieving the second highest reported score on CIFAR-100 (without additional pre-train data), second only to [7]. On Fashion-MNIST, Freiburg and CINIC10, to the best of our knowledge XNAS achieves a new state-of-the-art accuracy.

| **Datasets** Architecture | CIFAR100 Error | FMNIST Error | SVHN Error | Freiburg Error | CINIC10 Error | Params (M) | Search cost |
|---|---|---|---|---|---|---|---|
| Known SotA | **10.7** [7] | 3.65 [54] | **1.02** [7] | 10.7 [29] | 6.83 [29] | 26 [7] | - |
| PDARTS [6] | 15.9 | - | - | - | - | 3.6 | 0.3 |
| NAONet-1 [27] | 15.7 | - | - | - | - | 10.8 | 200 |
| NAONet-2 [27] | 14.7 | - | - | - | - | 128 | 200 |
| PDARTS-L [6] | 14.6 | - | - | - | - | 11 | 0.3 |
| SNAS[†] [50] | 16.5 | 3.72 | 1.98 | 14.7 | 7.13 | 2.8 | 1.5 |
| PNAS[†] [24] | 15.9 | 3.72 | 1.83 | 12.3 | 7.03 | 3.4 | 150 |
| Amoeba-A[†] [33] | 15.9 | 3.8 | 1.93 | 11.8 | 7.18 | 3.2 | 3150 |
| NASNet[†] [56] | 15.8 | 3.71 | 1.96 | 13.4 | 6.93 | 3.3 | 1800 |
| DARTS[†] [25] | 15.7 | 3.68 | 1.95 | 10.8 | 6.88 | 3.4 | 1 |
| ASAP[†] [29] | 15.6 | 3.73 | 1.81 | 10.7 | 6.83 | 2.5 | 0.2 |
| XNAS-Small | 13.6 | **3.64** | 1.72 | **6.3** | **6.0** | 3.7 | 0.3 |

Table 3: Classification errors of XNAS, compared to state-of-the-art NAS methods, on several datasets. Error refers to top-1 test error (%). The Search cost is measured in GPU days. Results marked with [†] are taken from [29], which tested and compared different cells on various datasets.

## 5 Related Work

Mentions of Experts in deep learning [32, 52, 11, 1] literature go decades back [18, 5], typically combining models as separable experts sub-models. A different concept of using multiple mixtures of experts as inner parts of a deep model, where each mixture has its own gating network, is presented in [10]. Following works build upon this idea and include a gating mechanism per mixture [51, 41], and some further suggest sparsity regularization over experts via the gating mechanism [37, 44]. These

gating mechanisms can be seen as a dynamic routing, which activates a single or a group of experts in the network on a per-example basis. Inspired by these works, our methods leverage PEA principled methods for automatically designing neural network inner components.

Furthermore, optimizers based on PEA theory may be useful for the neural architecture search phase. Common stochastic gradient-descent (SGD) and a set of PEA approaches, such as follow-the-regularized-leader (FTRL), were shown by [36, 14, 43] to be equivalent. Current NAS methods [56, 55, 30, 25, 2, 46, 23, 17, 29] use Adam, SGD with Momentum or other common optimizers. One notion that is common in PEA principled methods is the regret [4]. PEA strategies aim to guarantee a small regret under various conditions. We use the regret as a NAS objective, in order to establish a better principled optimizer than existing methods [23, 35]. Several gradient-descent based optimizers, such as Adam, present a regret bound analysis, however, the worst-case scenario for Adam has non-zero average regret [34], i.e., it is not robust. Our optimizer is designated for selecting architecture weights while achieving an optimal regret bound.

# 6   Conclusion

In this paper we presented XNAS, a PEA principled optimization method for differential neural architecture search. Inner network architecture weights that govern operations and connections, i.e. experts, are learned via exponentiated-gradient back-propagation update rule. XNAS optimization criterion is well suited for architecture-selection, since it minimizes the regret implied by sub-optimal selection of operations with tendency for sparsity, while enabling late bloomers experts to warm-up and take over during the search phase. Regret analysis suggests the use of multiple learning rates based on the amount of information carried by the backward gradient. A dynamic mechanism for wiping out weak experts is used, reducing the size of computational graph along the search phase, hence reducing the search time and increasing the final accuracy. XNAS shows strong performance on several image classification datasets, while being among the fastest existing NAS methods.

## Acknowledgements

We would like to thank the members of the Alibaba Israel Machine Vision Lab (AIMVL), in particular to Avi Mitrani, Avi Ben-Cohen, Yonathan Aflalo and Matan Protter for their feedbacks and productive discussions.

## Footnotes

[2]Experiments were performed using a NVIDIA GTX 1080Ti GPU.

[3]XNAS evaluation results can be reproduced using the code: https://github.com/NivNayman/XNAS

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
