[Supplementary Material]

# Supplementary Material

## 7 Proofs

### 7.1 Proof of Lemma 1

**Proof:** By contradiction, assume that expert $j$ is being wiped-out at the iteration $t$ based on line 12 in algorithm 1, and without loss of generality, $k \neq j$ is the leading expert at that time,

$$v_{t,k} = \arg\max_i \{v_{t,i}\}$$

$$v_{t,j} < v_{t,k} \cdot \exp\{-2\eta\mathcal{L}(T-t)\} \tag{7}$$

Since expert $j$ is the optimal one in hindsight, specifically,

$$v_{T,j} \geq v_{T,k} \tag{8}$$

However, since the loss is $\mathcal{L}$-bounded, the ratios between weights at time $t, T$ are bounded as well,

$$v_{T,k} \geq v_{t,k} \cdot \exp\{-\eta\mathcal{L}(T-t)\} \tag{9}$$

$$v_{T,j} \leq v_{t,j} \cdot \exp\{\eta\mathcal{L}(T-t)\} \tag{10}$$

Recap inequalities (8)-(10),

$$\begin{aligned}
v_{t,j} &\geq v_{T,j} \cdot \exp\{-\eta\mathcal{L}(T-t)\} \\
&\geq v_{T,k} \cdot \exp\{-\eta\mathcal{L}(T-t)\} \\
&\geq v_{t,k} \cdot \exp\{-2\eta\mathcal{L}(T-t)\}
\end{aligned}$$

In contradiction to (7). □

### 7.2 Proof of Lemma 2

**Proof: Left Hand Side.** Due to the non-negativity of $\{v_{t,i}\}_{i=1}^N$ for all $t = 1, \ldots, T$, we have,

$$\Gamma_t = 1 + \frac{\sum_{i\in I_{t-1}\setminus I_t} v_{t-1,i}}{\sum_{i\in I_t} v_{t,i}} \geq 1 \tag{11}$$

Hence,

$$\gamma_T = \prod_{t=1}^T \Gamma_t \geq \prod_{t=1}^T 1 = 1 \tag{12}$$

**Right Hand Side.** Denote $|\mathcal{S}|$ as the cardinal of the set $\mathcal{S}$. Then,

$$\begin{aligned}
\Gamma_t &= 1 + \frac{\sum_{i\in I_{t-1}\setminus I_t} v_{t,i}}{\sum_{i\in I_t} v_{t,i}} \\
&< 1 + \frac{|I_{t-1}\setminus I_t| \cdot \theta_t}{\sum_{i\in I_t} v_{t,i}} \tag{13} \\
&\leq 1 + \frac{|I_{t-1}\setminus I_t| \cdot \theta_t}{|I_t| \cdot \theta_t} \tag{14} \\
&= \frac{|I_t| + |I_{t-1}\setminus I_t|}{|I_t|} \\
&= \frac{|I_{t-1}|}{|I_t|}
\end{aligned}$$

where (13) is since $v_{t,i} < \theta_t \ \forall i \in I_{t-1}\setminus I_t$ and (14) is since $v_{t,i} \geq \theta_t \ \forall i \in I_t$, according to line 13 of algorithm 1. Thus taking the telescopic product yields,

$$\gamma_T = \prod_{t=1}^T \Gamma_t < \prod_{t=1}^T \frac{|I_{t-1}|}{|I_t|} = \frac{|I_0|}{|I_T|} = \frac{N}{|I_T|} \leq N \tag{15}$$

Finally we have,

$$1 \leq \gamma_T < N \tag{16}$$

□

## 7.3 Proof of XNAS Regret Bound

**Proof:** First let us state an auxiliary lemma,

**Lemma 3** (Hoeffding). *For any random variable $X$ with $\Pr(a \leq X \leq b) = 1$ and $s \in \mathbb{R}$,*

$$\ln \mathrm{E}\left[e^{sX}\right] \leq s\mathrm{E}\left[X\right] + \frac{s^2(b-a)^2}{8}. \tag{17}$$

Proof can be found in [15].

We start with defining experts' auxiliary and accumulated-auxiliary losses following [4] ,

$$\ell_t'(f_{t,i}) \quad = \quad \nabla\ell(p_t) \cdot f_{t,i} = -R_{t,i} \tag{18}$$

$$L_{T,i}' \quad = \quad \sum_{t=1}^{T} \ell_t'(f_{t,i}) \tag{19}$$

Notice that the auxiliary losses are bounded by an input parameter in line 1 of algorithm 1,

$$|\ell_t'(f_{t,i})| \leq \mathcal{L} \tag{20}$$

We also define the set of remaining experts at time $t$,

$$I_t = \{i | i \in I_{t-1} \text{ and } v_{t,i} \geq \theta_t\} \quad ; \quad I_0 = \{1, \ldots, N\} \tag{21}$$

Such that,

$$I_T \subset I_{T-1} \subset \cdots \subset I_0 \tag{22}$$

We Now bound the ratio of weights sums from both sides.
Let us derive a lower bound:

$$
\begin{aligned}
\ln \frac{V_T}{V_0} &= \ln \frac{\sum_{i \in I_T} v_{T,i}}{N} \\
&= \ln \left( \sum_{i \in I_T} \exp\left\{-\eta L_{T,i}'\right\} \right) - \ln N \\
&\geq \ln \max_{i \in I_T} \exp\left\{-\eta L_{T,i}'\right\} - \ln N \\
&= \ln \max_{i=1,\ldots,N} \exp\left\{-\eta L_{T,i}'\right\} - \ln N \\
&= -\eta \min_{i=1,\ldots,N} L_{T,i}' - \ln N
\end{aligned} \tag{23}
$$

Where we used lemma 1 in (23), assuring that loss minimizer is among the remaining experts.
Let us derive an upper bound:

$$
\begin{aligned}
\ln \frac{V_t}{V_{t-1}} &= \ln \frac{\sum_{i \in I_t} v_{t,i}}{\sum_{i \in I_{t-1}} v_{t-1,i}} \\
&= \ln \frac{\sum_{i \in I_{t-1}} v_{t,i}}{\sum_{i \in I_{t-1}} v_{t-1,i}} + \ln \frac{\sum_{i \in I_t} v_{t,i}}{\sum_{i \in I_{t-1}} v_{t,i}} \\
&= \ln \frac{\sum_{i \in I_{t-1}} v_{t,i}}{\sum_{i \in I_{t-1}} v_{t-1,i}} + \ln \frac{\sum_{i \in I_t} v_{t,i}}{\sum_{i \in I_t} v_{t,i} + \sum_{i \in I_{t-1} \backslash I_t} v_{t,i}} \\
&= \ln \frac{\sum_{i \in I_{t-1}} v_{t,i}}{\sum_{i \in I_{t-1}} v_{t-1,i}} - \ln \Gamma_t \\
&= \ln \frac{\sum_{i \in I_{t-1}} v_{t-1,i} \cdot \exp\left\{-\eta \ell_t'(f_{t,i})\right\}}{\sum_{i \in I_{t-1}} v_{t-1,i}} - \ln \Gamma_t \\
&\leq -\eta \frac{\sum_{i \in I_t} v_{t-1,i} \cdot \ell_t'(f_{t,i})}{\sum_{i \in I_t} v_{t-1,i}} + \frac{\eta^2 \mathcal{L}^2}{2} - \ln \Gamma_t \\
&= -\eta \ell_t'(p_t) + \frac{\eta^2 \mathcal{L}^2}{2} - \ln \Gamma_t
\end{aligned}
$$

$$\tag{24}$$
$$\tag{25}$$
$$\tag{26}$$
$$\tag{27}$$

Where (24) is due to (22) and (25) is by setting,

$$\Gamma_t := 1 + \frac{\sum_{i \in I_{t-1} \setminus I_t} v_{t,i}}{\sum_{i \in I_t} v_{t,i}} = \frac{\sum_{i \in I_t} v_{t,i} + \sum_{i \in I_{t-1} \setminus I_t} v_{t,i}}{\sum_{i \in I_t} v_{t,i}} \tag{28}$$

Inequality (26) results from lemma 3, with $[a, b] = [-\mathcal{L}, \mathcal{L}]$ and (27) is a result of the linearity of $\ell'_t$. Summing the logs telescopic sum:

$$\ln \frac{V_T}{V_0} = \sum_{t=1}^{T} \ln \frac{V_t}{V_{t-1}} \leq -\eta \sum_{t=1}^{T} \ell'_t(p_t) + \frac{\eta^2 T \mathcal{L}^2}{2} - \sum_{t=1}^{T} \ln \Gamma_t \tag{29}$$

Setting $\gamma_T := \prod_{t=1}^{T} \Gamma_t$ with bounds specified in by lemma 2, we have,

$$\ln \frac{V_T}{V_0} \leq -\eta \sum_{t=1}^{T} \ell'_t(p_t) + \frac{\eta^2 T \mathcal{L}^2}{2} - \ln \gamma_T \tag{30}$$

Combining the lower and upper bounds and dividing by $\eta$,

$$\sum_{t=1}^{T} \ell'_t(p_t) - \min_{i=1,\dots,N} L'_{T,i} \leq \frac{\eta T \mathcal{L}^2}{2} + \frac{1}{\eta} \ln N - \frac{1}{\eta} \ln \gamma_T \tag{31}$$

We now bound the accumulated regret, using the convexity of the loss,

$$\begin{aligned}
\text{Regret}_T &= \sum_{t=1}^{T} \ell_t(p_t) - \min_{i=1,\dots,N} \sum_{t=1}^{T} \ell_t(f_{t,i}) \\
&= \max_{i=1,\dots,N} \left\{ \sum_{t=1}^{T} \ell_t(p_t) - \ell_t(f_{t,i}) \right\} \\
&\leq \max_{i=1,\dots,N} \left\{ \sum_{t=1}^{T} \nabla \ell_t(p_t) \cdot (p_t - f_{t,i}) \right\} \\
&= \max_{i=1,\dots,N} \left\{ \sum_{t=1}^{T} \ell'_t(p_t) - \ell'_t(f_{t,i}) \right\} \\
&= \sum_{t=1}^{T} \ell'_t(p_t) - \min_{i=1,\dots,N} L'_{T,i}
\end{aligned} \tag{32}$$

Combining 31 and 32 completes the proof,

$$\text{Regret}_T \leq \frac{\eta T \mathcal{L}^2}{2} + \frac{1}{\eta} \ln N - \frac{1}{\eta} \ln \gamma_T \tag{33}$$

$\square$

# 8 Supporting Materials for the Key Properties Discussion

## 8.1 A Deterministic 3D Axes Toy Problem

In an attempt to demonstrate the possible recovery of *late bloomers*, we view an optimization problem in a three dimensional space as a prediction-with-experts problem, where each axis represents an expert with a constant prediction, i.e. $f_{t,x} \equiv (1, 0, 0)$, $f_{t,y} \equiv (0, 1, 0)$, $f_{t,z} \equiv (0, 0, 1)$ for the $x, y, z$ axes respectively. The forecaster then makes a prediction according to the following,

$$p_t = \frac{v_{t,x} \cdot f_{t,x} + v_{t,y} \cdot f_{t,y} + v_{t,z} \cdot f_{t,z}}{v_{t,x} + v_{t,y} + v_{t,z}} = x_t \cdot f_{t,x} + y_t \cdot f_{t,y} + z_t \cdot f_{t,z} = (x_t, y_t, z_t)$$

with $p_t$ in the simplex, i.e. $(x_t, y_t, z_t) \in \Delta = \{(x, y, z) \mid x, y, z \geq 0, x + y + z = 1\}$. Setting $v_{t,i} = \exp\{\alpha_{t,i}\}$ for $i \in \{x, y, z\}$, for a given loss function $\ell(p_t)$, the update terms for GD with softmax (section 3.2.1) and XNAS (section 2.2), associated with the $z$ axis, are as follows,

$$\textbf{(GD)} \; \partial_{\alpha_z}\ell(p_t) = z((x+y)\partial_z\ell(p_t) - x\partial_x\ell(p_t) - y\partial_y\ell(p_t)) \; \textbf{(XNAS)} \; \nabla_{p_t}\ell(p_t)f_{t,z} = \partial_z\ell(p_t)$$

as the corresponding terms for the $x$ and $y$ axes are similar by symmetry, see a full derivation in section 8.4.2.

Now let us present the following toy problem: A three dimensional linear loss function $\ell_1(x, y, z) = 2z$ is being optimized for 30 gradient steps over the simplex, i.e. $(x, y, z) \in \Delta$. Then for illustrating a region shift in terms of the best expert, the loss function shifts into another linear loss function $\ell_2(x, y, z) = -y - 2z$ which is then optimized for additional 90 steps. The trajectories are presented in Figure 3. At the first stage the $z$ axis suffers many penalties as its weight shrinks. Then it starts

Figure 3: Trajectories of XNAS and GD with softmax in 3D (left), projected on the simplex (middle) and the update terms for the $z$ axis (right). At the middle, color indicates progression through time, starting darker and ending lighter

receiving rewards. For GD with softmax, those rewards are attenuated, as explained in section 3.2.1 and can be seen in Figure 3 (right). Despite the linear loss, with constant gradients, the update term decays. Note that this effect is even more severe when dealing with more common losses of higher curvature, where the gradients decay near the local minimum and then further attenuated, as illustrated in section 8.2.4. Once the gradient shifts, it is already depressed due to past penalties, hence the $z$ axis struggles to recover. XNAS, however, is agnostic to the order of the loss values in time. Once the rewards balance out the penalties, the path leads towards the $z$ axis. In the meanwhile, the $y$ axis takes the lead.

## 8.2 A Deterministic 2D Axes Toy Problem

In section 8.1 we show the built-in attenuation of weak operations by GD with softmax. This is illustrated by a three dimensional toy example where the axes represent experts of constant predictions. Here we elaborate on this effect using a similar two dimensional toy problem, where the gradients with respect to the axes are the negative values of one another. See section 8.4.3 for the setup and full derivations. All the experiments in this section are conducted using a learning rate of $0.1$ for both optimizers for 50 steps.

### 8.2.1 Balanced Initialization

Figure 4 illustrates the attenuation of gradients for GD with softmax, as although the gradients of the loss are constant, the gradients' magnitude decreases as we move away from the initialization $\alpha_x = \alpha_y$, i.e. $(x, y) = (0.5, 0.5)$. XNAS indeed receives constant gradients thus reaches the minima faster.

### 8.2.2 Imbalanced Initialization

The attenuated gradients also make GD with softmax more sensitive to the initialization, as demonstrated in Figure 5, where $\alpha_x = 0 < 5 = \alpha_y$ and GD with softmax, whose gradients are attenuated, makes no progress while XNAS reaches the minima.

Figure 4: Trajectories of GD with softmax and XNAS for a $\ell(x, y) = -x$ (left) the distance from the optimal point (middle) and the corresponding magnitude of the gradients (right) for a balanced initialization $\alpha_y = \alpha_x$. The vertical lines are the contour lines of the loss function. The dashed line is the simplex $x + y = 1$, and the trajectories are the solid lines with circles at their ends.

Figure 5: Trajectories of GD with softmax and XNAS for a $\ell(x, y) = -x$ (left) the distance from the optimal point (middle) and the corresponding magnitude of the gradients (right) for an imbalanced initialization $\alpha_x < \alpha_y$. The vertical lines are the contour lines of the loss function. The dashed line is the simplex $x + y = 1$, and the trajectories are the solid lines with circles at their ends.

### 8.2.3 The Preference of Dense Solutions

Presenting the attenuation factor $x \cdot y$ on the simplex, i.e. $x \cdot (1 - x)$, in Figure 6, demonstrates how gradients are harder attenuated as far away as the variables move from a dense solution, e.g. $(x, y) = (0.5, 0.5)$ at $\alpha_x = \alpha_y$. Hence, it is harder for GD with softmax to strengthen a single expert

Figure 6: The attenuation factor of GD with softmax along the simplex (left) and versus $\alpha_x$ as $\alpha_y = 0$ (right)

over the other. This effect encourages dense solutions over sparse solutions, i.e. a choice of a single expert. Due to the *discretization stage*, described in section 2.1, the denser the solution is, the more degradation in performance is incurred. Hence dense solutions should be discouraged rather than encouraged.

### 8.2.4 Loss Functions of a Higher Curvature

Sections 8.2.1 and 8.2.2 deal with a linear loss function of no curvature and constant gradients. Once convex loss functions are considered, the gradients decrease towards the local minimum. Figure 7 illustrates the further attenuation of GD with softmax for a quadratic loss function $\ell(x, y) = 0.5(x - 1)^2 + 0.5y^2$, which makes it even harder for GD with softmax to reach the local minimum.

Figure 7: Trajectories of GD with softmax and XNAS for a $\ell(x, y) = 0.5(x - 1)^2 + 0.5y^2$ (left) the distance from the optimal point (middle) and the corresponding magnitude of the gradients (right). The circular lines are the contour lines of the loss function. The dashed line is the simplex $x + y = 1$, and the trajectories are the solid lines with circles at their ends.

## 8.3 Regret and Correct Selection in Statistical Setting

Figure 8: Correct selection fraction and regret of optimizers.

We consider a statistical setup for comparing XNAS with the common Gradient Descent (GD) with softmax, described in section 3.2.1. This setup simulates the iterative architecture optimization and final selection of the top expert for a single forecaster.

Two forecasters are compared, XNAS and GD with softmax. Both receive noisy independent and identically distributed (i.i.d) rewards of $N$ experts. Each expert has an initial i.i.d bias $\{b_i\}_{i=1}^{N} \sim$ N$(0, 1)$ simulating its inherent value, so its rewards satisfy $R_{t,i} \sim$ N$(b_i, \sigma_R^2)$ for $i = 1, \ldots, N$ and $t = 1, \ldots, T$, where N$(\mu, \sigma^2)$ is a Gaussian distribution with a mean $\mu$ and a standard deviation $\sigma$.

The first forecaster updates its weights using GD with softmax update rule from Equation 6 (full derivation in section 8.4.1), common to previous NAS methods, while the second is using Algorithm 1.

The forecasters use their update rules to update weights along the run. At the end of the run each selects the expert with the largest weight. A correct classification satisfies $\max\limits_{i=1,\ldots,N} \alpha_i = \max\limits_{i=1,\ldots,N} b_i$.

The average regret of those runs is also calculated based on equation 2.

Figure 8 shows a mean of 1000 Monte-Carlo runs, each of 1000 time-steps, plotting the regret and the fraction of correct selection (classification). In Figure 8 (left), both terms are plotted versus a varying number of experts. It can be seen that the regret of XNAS is significantly smaller, scaling

with the number of experts like $O(\sqrt{\ln N})$, as implied by its regret upper bound in equation 4, while GD regret scales like $O(\sqrt{N})$ [14].

In Figure 8 (right), the noise standard deviation $\sigma_R$ is varying, making it harder to correctly classify the expert with the highest bias. Again, XNAS dominates GD with softmax, which is more sensitive to the noisy rewards due to the 'late bloomers' described in 3.2.1, e.g. the best experts might suffer some large penalties right at the beginning due to the noise, thus might not recover for GD with softmax.

In both graphs it can be seen that the correct selection fraction is monotonically decreasing as the regret is increasing. This gives an additional motivation for the use of the regret minimization approach as a criterion for neural architecture search.

## 8.4 Gradients Derivations

For the comparison to previous work [25], we consider the decision variables $\alpha_{t,i} = \ln v_{t,i}$, as the right hand side is defined at (1).

### 8.4.1 The Derivation of Derivatives in the General Case

$$
\begin{aligned}
\frac{\partial u_{t,j}}{\partial \alpha_{t,i}} &= \frac{\partial}{\partial \alpha_{t,i}} \frac{e^{\alpha_{t,j}}}{\sum_{k=1}^{N} e^{\alpha_{t,k}}} \\
&= \frac{\delta_{i,j} e^{\alpha_{t,i}} \cdot \sum_{k=1}^{N} e^{\alpha_{t,k}} - e^{\alpha_{t,j}} \cdot e^{\alpha_{t,i}}}{\left( \sum_{k=1}^{N} e^{\alpha_{t,k}} \right)^2} \\
&= \frac{e^{\alpha_{t,i}}}{\sum_{k=1}^{N} e^{\alpha_{t,k}}} \left( \delta_{i,j} - \frac{e^{\alpha_{t,j}}}{\sum_{k=1}^{N} e^{\alpha_{t,k}}} \right) \\
&= u_{t,i} \left( \delta_{i,j} - u_{t,j} \right)
\end{aligned}
\tag{34}
$$

where $\delta_{i,j} = \begin{cases} 1, & \text{if } i = j, \\ 0, & \text{if } i \neq j. \end{cases}$ is the Kronecker delta.

Observe that:

$$
\begin{aligned}
\frac{\partial \ell_t(p_t)}{\partial u_{t,i}} &= \nabla_{p_t} \ell_t(p_t)^T \cdot \frac{\partial p_t}{\partial u_{t,i}} \\
&= \nabla_{p_t} \ell_t(p_t)^T \cdot \frac{\partial}{\partial u_{t,i}} \sum_{j=1}^{N} u_{t,j} f_{t,j} \\
&= \nabla_{p_t} \ell_t(p_t)^T \cdot f_{t,i}
\end{aligned}
\tag{35}
$$

Finally,

$$
\begin{aligned}
\frac{\partial \ell_t(p_t)}{\partial \alpha_{t,i}} &= \sum_{j=1}^{N} \frac{\partial \ell_t(p_t)}{\partial u_{t,j}} \cdot \frac{\partial u_{t,j}}{\partial \alpha_{t,i}} \\
&= \sum_{j=1}^{N} \nabla_{p_t} \ell_t(p_t)^T \cdot f_{t,j} \cdot u_{t,i} \left( \delta_{i,j} - u_{t,j} \right) \\
&= \nabla_{p_t} \ell_t(p_t)^T \cdot u_{t,i} \left( f_{t,i} - \sum_{j=1}^{N} u_{t,j} f_{t,j} \right) \\
&= \nabla_{p_t} \ell_t(p_t)^T \cdot u_{t,i} \left( f_{t,i} - p_t \right)
\end{aligned}
\tag{36}
$$
$$
\tag{37}
$$

where (36) is due to (34) and (35).

### 8.4.2 The Derivation of Derivatives for the 3D Axes Problem

In this section we derive the derivatives with respect to the $x, y, z$ axes for the toy problem introduced at section 8.1. In this case,

$$
\begin{aligned}
f_{t,x} &\equiv (1,0,0)^T & u_{t,x} &= x_t \\
f_{t,y} &\equiv (0,1,0)^T & u_{t,y} &= y_t \\
f_{t,z} &\equiv (0,0,1)^T & u_{t,z} &= z_t
\end{aligned}
\tag{38}
$$

Hence,

$$
p_t = \sum_{i \in \{x,y,z\}} u_{t,i} f_{t,i} = x_t(1,0,0)^T + y_t(0,1,0)^T + z_t(0,0,1)^T = (x_t, y_t, z_t)^T \tag{39}
$$

such that for a loss function $\ell$,

$$
\nabla_{p_t} \ell(p_t) = \left( \frac{\partial \ell(p_t)(p_t)}{\partial x_t}, \frac{\partial \ell(p_t)(p_t)}{\partial y_t}, \frac{\partial \ell(p_t)(p_t)}{\partial z_t} \right)^T = (\partial_{x_t}\ell(p_t), \partial_{y_t}\ell(p_t), \partial_{z_t}\ell(p_t))^T \tag{40}
$$

Then, setting (38) and (40) in (37), we have,

$$
\begin{aligned}
\partial_{\alpha_x}\ell(p_t) = \frac{\partial \ell(p_t)}{\partial \alpha_x} &= (\partial_{x_t}\ell(p_t), \partial_{y_t}\ell(p_t), \partial_{z_t}\ell(p_t))^T \cdot x_t \left( (1,0,0)^T - (x_t, y_t, z_t)^T \right) \\
&= (\partial_{x_t}\ell(p_t), \partial_{y_t}\ell(p_t), \partial_{z_t}\ell(p_t))^T \cdot x_t (y_t + z_t, -y_t, -z_t)^T \tag{41} \\
&= x_t \left( (y_t + z_t)\partial_{\alpha_x}\ell(p_t) - y_t\partial_{\alpha_y}\ell(p_t) - z_t\partial_{\alpha_z}\ell(p_t) \right) \tag{42}
\end{aligned}
$$

where (41) is since $(x, y, z) \in \Delta$, defined in section 8.1. By symmetry we have,

$$
\partial_{\alpha_y}\ell(p_t) = y_t \left( (x_t + z_t)\partial_{\alpha_y}\ell(p_t) - x_t\partial_{\alpha_x}\ell(p_t) - z_t\partial_{\alpha_z}\ell(p_t) \right) \tag{43}
$$

$$
\partial_{\alpha_z}\ell(p_t) = z_t \left( (x_t + y_t)\partial_{\alpha_z}\ell(p_t) - x_t\partial_{\alpha_x}\ell(p_t) - y_t\partial_{\alpha_y}\ell(p_t) \right) \tag{44}
$$

The update terms for XNAS are according to (35),

$$
\nabla_{p_t}\ell(p_t)^T \cdot f_{t,x} = (\partial_{x_t}\ell(p_t), \partial_{y_t}\ell(p_t), \partial_{z_t}\ell(p_t)) \cdot (1,0,0)^T = \partial_{x_t}\ell(p_t) \tag{45}
$$

$$
\nabla_{p_t}\ell(p_t)^T \cdot f_{t,y} = (\partial_{x_t}\ell(p_t), \partial_{y_t}\ell(p_t), \partial_{z_t}\ell(p_t)) \cdot (0,1,0)^T = \partial_{y_t}\ell(p_t) \tag{46}
$$

$$
\nabla_{p_t}\ell(p_t)^T \cdot f_{t,z} = (\partial_{x_t}\ell(p_t), \partial_{y_t}\ell(p_t), \partial_{z_t}\ell(p_t)) \cdot (0,0,1)^T = \partial_{z_t}\ell(p_t) \tag{47}
$$

### 8.4.3 The Derivation of Derivatives for a 2D Axes Problem

In this section we derive the derivatives with respect to the $x, y$ axes for a two dimensional toy problem. Similar to section 8.1 where a three dimensional problem was considered, now we consider only two axes. Each axis represents an expert of a constant prediction,

$$
\begin{aligned}
f_{t,x} &\equiv (1,0)^T & u_{t,x} &= x_t \\
f_{t,y} &\equiv (0,1)^T & u_{t,y} &= y_t
\end{aligned}
\tag{48}
$$

Where $x_t + y_t \equiv 1$. Hence,

$$
p_t = \sum_{i \in \{x,y\}} u_{t,i} f_{t,i} = x_t(1,0)^T + y_t(0,1)^T = (x_t, y_t)^T \tag{49}
$$

such that for a loss function $\ell$,

$$
\nabla_{p_t}\ell(p_t) = \left( \frac{\partial \ell(p_t)}{\partial x_t}, \frac{\partial \ell(p_t)}{\partial y_t} \right)^T = (\partial_{x_t}\ell(p_t), \partial_{y_t}\ell(p_t))^T \tag{50}
$$

Then, setting (48) and (50) in (37), we have,

$$
\begin{aligned}
\partial_{\alpha_x}\ell(p_t) = \frac{\partial \ell(p_t)}{\partial \alpha_x} &= (\partial_{x_t}\ell(p_t), \partial_{y_t}\ell(p_t))^T \cdot x_t \left( (1,0)^T - (x_t, y_t)^T \right) \\
&= (\partial_{x_t}\ell(p_t), \partial_{y_t}\ell(p_t))^T \cdot x_t (y_t, -y_t)^T \tag{51} \\
&= x_t y_t (\partial_{x_t}\ell(p_t) - \partial_{y_t}\ell(p_t)) \tag{52}
\end{aligned}
$$

where (51) is since $x_t + y_t \equiv 1$. By symmetry we have,

$$
\partial_{\alpha_y}\ell(p_t) = x_t y_t (\partial_{y_t}\ell(p_t) - \partial_{x_t}\ell(p_t)) = -\partial_{\alpha_x}\ell(p_t) \tag{53}
$$

## 8.5 The Mean Normalized Entropy

In this section we provide the technical calculation details of the mean normalized entropy, referred to in section 3.2.2. The normalized entropy of forcaster $(i, j)$ is calculated at the end of the search as following,

$$\bar{H}_T^{(i,j)} = -\frac{1}{\ln(N)} \sum_{i=1}^{N} u_{T,i}^{(i,j)} \ln\left(u_{T,i}^{(i,j)}\right) \tag{54}$$

The mean is taken over all the forecasters in a normal cell, i.e.

$$\bar{H}_T = \frac{1}{|\mathcal{I}| \cdot |\mathcal{J}|} \sum_{i \in \mathcal{I}} \sum_{j \in \mathcal{J}} \bar{H}_T^{(i,j)} \tag{55}$$

where $\mathcal{I}$ and $\mathcal{J}$ are the sets of indices $i$ and $j$ respectively.

# 9 Detailed Experiments Setting

## 9.1 Classification Datasets Details

In this section we will describe the additional datasets that were used for transferability tests in section 4.3

**CINIC-10: [8]** is an extension of CIFAR-10 by ImageNet images, down-sampled to match the image size of CIFAR-10. It has $270,000$ images of 10 classes, i.e. it has larger train and test sets than those of CIFAR-10.

**CIFAR-100: [42]** A natural image classification dataset, containing 100 classes with 600 images per class. The image size is 32x32 and the train-test split is $50,000{:}10,000$ images respectively.

**FREIBURG: [19]** A groceries classification dataset consisting of 5000 images of size 256x256, divided into 25 categories. It has imbalanced class sizes ranging from 97 to 370 images per class. Images were taken in various aspect ratios and padded to squares.

**SVHN: [28]** A dataset containing real-world images of digits and numbers in natural scenes. It consists of $600,000$ images of size 32x32, divided into 10 classes. The dataset can be thought of as a real-world alternative to MNIST, with an order of magnitude more images and significantly harder real-world scenarios.

**FMNIST: [47]** A clothes classification dataset with a $60,000{:}10,000$ train-test split. Each example is a grayscale image of size 28x28, associated with a label from 10 classes of clothes. It is intended to serve as a direct drop-in replacement for the original MNIST dataset as a benchmark for machine learning algorithms.

## 9.2 CIFAR-10 XNAS Search details

**Data pre-processing.** We apply the following:

- Centrally padding the training images to a size of 40x40.
- Randomly cropping back to the size of 32x32.
- Randomly flipping the training images horizontally.
- Auto augment.
- Standardizing the train and validation sets to be of a zero-mean and a unit variance.

**Operations and cells.** We select from the operations mentioned in 4.1, used with stride 1 on normal cells, and with stride 2 on reduction cells in edges connected to the two previous cells. Other edges in reduction cells are used with stride 1. Convolutional layers are padded so that the spatial resolution is kept. The operations are applied in the order of ReLU-Conv-BN. Following [29],[33], depthwise separable convolutions are always applied twice. The cell's output is a 1x1 convolutional layer applied on all of the cells' four intermediate nodes' outputs concatenated, such that the number of channels is preserved. In CIFAR-10, the search lasts up to 0.3 days on NVIDIA GTX 1080Ti GPU.

## 9.3 Train Details

**CIFAR-10.** The training architecture consists of stacking up 20 cells: 18 normal cells and 2 reduction cells, located at the $1/3$ and $2/3$ of the total network depth respectively. For the three architectures XNAS-Small, XNAS-Medium and XNAS-Large, the normal cells start with 36, 44 and 50 channels respectively, where we double the number of channels after each reduction cell. We trained the network for 1500 epochs using a batch size of 96 and SGD optimizer with nesterov-momentum of 0.9. Our learning rate regime was composed of 5 cycles of power cosine annealing learning rate [17], with amplitude decay factor of 0.5 per cycle and initial value of 0.025. For regularization we used cutout [9] with a length of 16, scheduled drop-path [22] of 0.2, auxiliary towers [39] after the last reduction cell with a weight of 0.4, label smoothing [40] of 0.1, AutoAugment [7] and weight decay of $3 \cdot 10^{-4}$.

**ImageNet.** Our training architecture starts with stem cells that reduce the input image resolution from 224 to 56 (3 reductions), similar to [25]. We then stack 14 cells: 12 normal cells and 2 reduction cells. The reduction cells are placed after the fourth and eighth normal cells. The normal cells start with 46 channels, as the number of channels is doubled after each reduction cell. For each cell we also add a SE layer [16]. We trained the network, with a batch size of 1280 for 250 epochs, with one cycle of power cosine learning rate, weight decay of $10^{-4}$ and nesterov-momentum of 0.9. We add an auxiliary loss after the last reduction cell with a weight of 0.4. During training, we normalize the input image and crop it with a random cropping factor in the range of 0.08 to 1. In addition we use auto-augment and horizontal flipping. During testing, we resize the input image to the size of 256x256 and applying a fixed central crop to the size of 224x224.

**Additional datasets.** Our additional classification datasets consist of CINIC-10 [8], CIFAR-100 [42], FREIBURG [19], SVHN [28] and FashionMNIST [47]. Their training scheme was similar to the one used for CIFAR-10, described at 9.3, with some minor adjustments and modifications. For the FREIBURG dataset, we resized the original images from 256x256 to 96x96 and used a batch size of 16. For CINIC-10, we trained the network for 800 epochs instead of 1500, since this dataset is much larger then CIFAR-10. For Fashion-MNIST we edited the learned augmentations regime to fit a dataset of grayscale images.

## 10 Evaluation Details

### 10.1 Cells Learned by XNAS

Figure 9 presents the cells learned by XNAS on CIFAR-10.

Figure 9: XNAS learned normal and reduction cells on CIFAR-10.

**Cell depth:** When comparing the cell in Figure 9 to other reported NAS cells [29, 25, 33, 56, 50, 24], it is clear visually that XNAS cell is "deeper" in some sense.

We wish to define a metrics for a cell "depth". A cell $C_k$ contains four nodes. Each node can be connected to the previous cell's nodes or to the two previous cells' outputs $C_{k-2}$, $C_{k-1}$. Let us index the previous cells $C_{k-2}$, $C_{k-1}$ and the cell's four nodes as $0, \ldots, 5$ respectively.

Define the depth of each connection in a cell as the index of the node (or previous cell) it came from. A simple metric for a cell depth can be the average depth of its inner connections. Table 4 presents the depth of XNAS and other NAS methods normal cells.

| Cell | Depth |
|------|-------|
| SNAS [50] | 0.625 |
| PNAS [24] | 0.5 |
| Amoeba-A [33] | 0.9 |
| NASNet [56] | 0.4 |
| DARTS [25] | 0.625 |
| ASAP [29] | 0.875 |
| XNAS | 1.375 |

Table 4: XNAS and other NAS methods normal cell Depth.

We can see from Table 4 that the XNAS cell is much deeper then a typical NAS cell. This observation could provide a hint about the superior performance of the XNAS cell. Unlike most NAS methods, that usually produce shallow cells, XNAS cell utilizes better the possible degree of freedom of a NAS cell design, yielding a deeper and more complex architecture.

## 10.2 Flops and Inference Time

The number of flops and the inference time are also measured for the XNAS models:

| CIFAR-10 Architecture | Test Error [%] | Parameters [M] | Flops [M] | Inference Time [ms] |
|------------------------|----------------|----------------|-----------|---------------------|
| XNAS-Small | 1.81 | 3.7 | 621 | 1.98 |
| XNAS-Medium | 1.73 | 5.6 | 905 | 1.99 |
| XNAS-Large | 1.60 | 7.2 | 1150 | 1.94 |

Table 5: Classification errors, number of parameters, number of flops and inference time per sample of XNAS models on CIFAR-10. Test error refers to top-1 test error (%). XNAS Small, Medium and Large refer to network architectures with XNAS cells and 36, 44 and 50 initial channels respectively.

| ImageNet Architecture | Test Error [%] | Parameters [M] | Flops [M] | Inference Time [ms] |
|------------------------|----------------|----------------|-----------|---------------------|
| XNAS | 23.9 | 5.2 | 590 | 1.49 |

Table 6: Transferability classification error, number of parameters, number of flops and inference time per sample of XNAS models on ImageNet. Test error refers to top-1 test error (%). See the model details in section 4.3.