[Reviews · NeurIPS 2019]

Reviewer 1



Pro: +It provides a better theoretical guidance for learning the hyper parameters for model selection using the PEA theory, which accelerates DARTS . +The results are impressive, though conjunction with a bunch of data augmentation methods. + Theoretical proofs are valid. Cons: - The paper is somehow difficult to read, check the comments below. It needs clearer description. One thing for CIFAR10 is the author adopted many tricks (275 page 7) for learning. However, some of the works such as DARTS do not use part of it such as auto augmentation etc. Baseline algorithms should add same tricks for fair comparison. There are two proposals that the author proposed, 1) regret for evaluation 2) wipeout. I think it also worths to see how different components affects the results in ablation rather than simply showing the final results. Also ablation about the model FLOPs or inference time are valuable Comments of the paper. The author us (#) as citation, which save space, but raise confusion between equation and citation (especially in checking the proof at supplementary). Change () to [] for better distinguish the two. In PEA setting, there are p_t for prediction from the network. p^(j, k) as prediction from edge. I found this is confusing, better clarify the two p are predicting different things. Bigger problem is at Fig.1, the visualization is not clear, it should have better notation with correspondence to PEA setting and the fonts are too small for a clear understanding of the figure. Make it better and clearer is a must. Please relocate Fig.1 and All.1 to Page 3 for better readability. ================================================== The author's feedback solved my concern, I keep the score the same. However, it also needs better clarification about the holding of assumptions.

Reviewer 2



The paper proposes a new optimization algorithm for differentiable architecture search. The main difference compared to other differentiable methods is that it adds a wipe-out step, which eliminates the "experts" that have very small scores. The paper is not clearly written. It seems to be that the formulation of the paper is very similar to DARTS. It changes the name of "component" in DARTS to experts, but it does not make a fundamental difference. In terms of optimization algorithm, the main difference compared to DARTS method is that it has a wipe-out method. I think the main benefits of the wipe out is the search speed, but it has the risk of eliminating useful "expert" at the beginning. Not all the notations in Algorithm 1 are well defined. This makes understanding of the algorithm difficult. In general, I do not see enough novelty in the proposed method, and the writing is not clear enough.

Reviewer 3



[Based on the author feedback, I have upgraded my score. I think the empirical evidence for exponentiated gradient descent (+wipeout) in NAS is valuable for the community. I would suggest that the authors clearly state the limitations associated with the analysis.] Summary: The proposed approach for NAS treats architecture choices, i.e., intra-cell node connections as in DARTS (Liu et al., 2018), as selection among "experts". The expert weights are found via EG descent (Kivinen & Warmuth, 1997) that somehow utilizes the standard "back-propagated loss gradient" (line 113) to perform the multiplicative weight updates. It's at this point that I had trouble following the theoretical analysis given that the EG algorithm requires a loss function on the expert prediction to be specified for the development of a regret bound. The paper does not define this loss function (\ell_s() in (2)), and it isn't immediately clear to me what it could be in this context: what is the loss at the output of an intermediate node of a cell in the middle of a parent network during search? Further, the requirement in the main result that this loss be convex in its prediction makes me wonder whether the regret analysis actually holds for the setting of the paper. The experimental results, however, appear to be good enough to at least warrant further investigation into these kinds of multiplicative weight updates. Specifically, the speed of search is fast owing to either the EG descent algorithm, the proposed wipeout operation, or some combination. Originality: I can't speak to the application of EG to training neural networks generally, but I thought it was a clever idea in the context of NAS where hard selections among finite sets must be made. Quality: For the main result, a convex loss in the forecaster prediction is required. However, the paper also states that explicit loss is not assigned (line 113). I may be missing something simple, but I don't see how the regret bound holds. Can the authors provide the form of this loss function? The experimental setup leading to the performance comparisons provided in the tables of Section 5 appears to be fair, but this reviewer is not 100% certain. How much of the speed-up is due to the use of EG vs. EG+wipeout? (It would have been nice to see what the experimental performance would have been without the wipeout step.) Clarity: Both the regret bound itself, and the determination of an optimal learning rate require an absolute bound on the loss. What was this set to for the experiments? Also, I don't understand the relevance of the note in line 228 regarding the gradient clipping value. How is this important?

[Author Response · NeurIPS 2019]

**Reviewer #2**: Thank you for the thorough review. Indeed, better theoretical guidance enables XNAS to naturally select several hyper parameters, achieving best performance and accelerate differential architecture search.

Experiments: since the compared methods use different sets of training tricks, we adopted the exact ones reported in SHARPDARTS, which is the newest NAS compared. The additional tricks in SHARPDARTS vs DARTS are: 1.AutoAugment 2.power cosine annealing learning rate, instead of cosine annealing.

Since DARTS' code is published, a fair comparison is attainable:

| CIFAR-10 error | DARTS repo | XNAS repo | FLOPS(M) | inference time (ms, 1080ti) |
|---|---|---|---|---|
| DARTS-V2 (36 channels) | 2.76% | **2.42**% | 550 | 4.2 |
| XNAS-small (36 channels) | **2.16**% | 1.81% | 529 | 3.6 |

In Bold are the results of new experiments following the reviewer's request. It can be seen that while additional tricks do help, XNAS cells exhibit significantly better performance even without those. For clarity, we will add 'XNAS-plain' results in the camera-ready version. Plain-mode is now supported by XNAS repo, which will be published upon acceptance to enable further comparisons. FLOPs and inference time comparisons were out of this paper's scope, and were not reported by the majority of compared methods. We will report our results in the revised experiments section.

EG/Wipeout ablation: Please find EG/Wipeout ablation below.

Paper clarity: the revised version now includes clearer explanations and modified figures as remarked by the reviewer.

Specifically, $p_t \doteq p_t^{(j,k)}$ is the prediction of a forecaster, as "superscript indices are ignored for brevity" (line 90).

For citation we simply used the command *cite* with NeurIPS style.

Fig.1 was replaced with an illustration of multiplicative vs additive updates for differential architecture search.

**Reviewer #3**: Thank you for your review. Apparently our contribution and novelty was not stated clearly.

*main difference compared to DARTS method is that it has a wipe-out method* - While DARTS picks a generic optimizer for architecture optimization, we derive an optimizer tailor-made for NAS based on EG. We suggest regret-minimization as a novel criterion for NAS to derive XNAS, and prove it achieves optimality under this criterion. We explore the algorithm's appealing properties, like robustness to initialization and the usage of multiple theoretically-derived learning rates, and demonstrate these in different setups. XNAS is lighter and x3.3 faster than DARTS, thanks to simple multiplicative updates and less hyper-parameters involved. It outperforms DARTS in $7/7$ datasets, including $-42\%$ error in CIFAR-10. The wipeout is merely a part of this synergy. Please find EG/Wipeout ablation below.

*wipeout ... has the risk of eliminating useful "expert"* - the wipeout's merit is the elimination of unuseful experts only. Lemma 1 states that clearly: "In XNAS, The optimal expert in hindsight cannot be wiped-out".

In the light of the above, we ask the reviewer to reconsider his review for this work.

**Reviewer #4**: Thank you for the valuable remarks which help us make the revised version much clearer, and also for acknowledging the originality and the need for EG in the context of NAS.

Regret analysis assumptions: *provided under conditions that are not obvious to hold* - that is correct. In the analysis, we assume gradient-boundness and convexity, as the former is enforced by applying gradient-clipping, the latter does not hold in general for DNNs. However, the convexity of the loss is a very common assumption for regret minimization and online learning in general. It is assumed when deriving similar theoretical guarantees for *all* previous NAS optimizers, e.g. gradient descent (ASAP) and ADAM (DARTS). In addition, we put to test the optimizer in several toy setups and simulations, deterministic and statistical (sections 8.1, 8.2, 8.3), investigate its properties and compare with common optimizers. Finally, extensive experimentation in NAS provides an empirical support for the algorithm derived under that assumption. Based on that, we believe that XNAS analysis has a high significance as well.

EG loss function: *Can the authors provide the form of this loss function?* - The loss function is the parent network's cross-entropy loss over the validation set. This loss is not defined explicitly for a predictor in an intermediate node. However, its back-propagated gradient is: $\nabla_{p_t}\ell_{\text{val}}(p_t)$. Therefore, unlike loss-based PEA methods like HEDGE, we derived a gradient-based NAS optimizer, via auxiliary-losses (Eq. 22). We will make this observation clearer.

Reward bounds, gradient clipping and optimal learning rates - the bounds are over the reward values (Alg.1, line 8), $|R_{t,i}| \leq \mathcal{L}$, as required by our theory and fulfilled by a simple gradient-clipping. We will make that clearer in the paper. The optimal learning rate derived from theorem 1 depends on that bound (line 163) and is used in our experiments (values-line 229). The successful usage of the theoretical learning rate in practice is a key property of XNAS.

EG/Wipeout ablation: The revised version will contain a section for the analysis of EG without wipeout. We include here only the summary and omit the full details and graphs due to the limited space.

Runtime: *EG*: 0.35 GPU-days. Faster than ADAM which is used in DARTS(1nd) and runs for 0.4 GPU-days with a significantly inferior performance (table 2). This is mostly due to fewer calculations and no momentum updates (section 4.2.2). *Wipeout contribution*: $-0.05$ GPU-days. Wipeout takes effect around the last $30\%$ of the run, and reduces this part's runtime by half. If one desires a further acceleration (e.g. a large dataset or many operations), while risking with the elimination of useful experts - the wipeout factor discussed in section 4.2.1 controls this trade-off.

Performance: EG is responsible for most of the improvement compared to previous methods. Based on a few runs, the mean error for CIFAR-10 is around $1.75\%$ (50 channels). The wipeout contributes to the overall mean performance, and also reduces its standard deviation caused by hard selections of operations ('relaxation bias', line 128).

[Meta-Review · NeurIPS 2019]

The reviewers appreciated the good empirical results and theoretical analysis. This paper proposes to treat NAS problems as a selection problem among experts. Over time, it eliminates underperforming experts with a wipe-out step. As two of the reviewers pointed out, the theoretical analysis is interesting (and rare, in this type of paper), but it would be good to more explicitly spell out when and why the assumptions hold. Empirical performance seems good, but the authors should include error bars for at least the CIFAR-10 experiments and ideally the ImageNet ones as well. The architecture search also seems to be very fast compared to other methods (e.g. DARTS), but again, it would be good to more clearly spell out if this was done over multiple seeds and if the cost per seed is included in the total cost of the search (as done in the DARTS paper). Overall, this is a good paper. Two reviewers argued for acceptance, while the last reviewer gave a relatively low score. The rebuttal didn't sway this reviewer towards changing their score, but they recognized that the algorithm proposed in this paper is interesting and has practical value. As a result, they chose not to directly argue against acceptance and stated they would not be upset if the paper was accepted.